# Utilizing Machine Learning for Detecting Harmful Situations by Audio and Text

Merav Allouch [1], Noa Mansbach [1], Amos Azaria [1] and Rina Azoulay [2,*]

1   Department of Computer Science, Ariel University, Ariel 40700, Israel
2   Department of Computer Science, Jerusalem College of Technology, Jerusalem 9372115, Israel
*   Correspondence: azrina@g.jct.ac.il

**Abstract:** Children with special needs may struggle to identify uncomfortable and unsafe situations. In this study, we aimed at developing an automated system that can detect such situations based on audio and text cues to encourage children's safety and prevent situations of violence toward them. We composed a text and audio database with over 1891 sentences extracted from videos presenting real-world situations, and categorized them into three classes: neutral sentences, insulting sentences, and sentences indicating unsafe conditions. We compared insulting and unsafe sentence-detection abilities of various machine-learning methods. In particular, we found that a deep neural network that accepts the text embedding vectors of bidirectional encoder representations from transformers (BERT) and audio embedding vectors of Wav2Vec as input attains the highest accuracy in detecting unsafe and insulting situations. Our results indicate that it may be applicable to build an automated agent that can detect unsafe and unpleasant situations that children with special needs may encounter, given the dialogue contexts conducted with these children.

**Keywords:** text classification; audio classification; machine learning; bulling; pretrained models; children's safety; assistive technologies for persons with disabilities

## 1. Introduction

The ability of identifying threatening and socially challenging situations is crucial for all humans, particularly for children and especially children with special needs. Children with autism spectrum disorder (ASD) may struggle to identify these situations and encounter difficulties in communicating with other people. These children may be involved in unsafe situations without being aware of it. Therefore, the development of an AI-based system for assisting the special needs children in handling such situations may be very useful. The system could enable early intervention, ensuring safety and preventing harm.

To demonstrate the importance of such an automated system, consider a child with special needs entering an unfamiliar environment with other children. This child may feel scared and may unintentionally ask embarrassing questions. Additionally, the child may be insulted by other children or even be targeted by individuals seeking to exploit the child's vulnerabilities. An AI-based system may intervene and minimize the impact of such harmful situations.

While there have been many attempts to use AI for assisting children with special needs, many of these efforts focus on enhancing specific skills or abilities of the special needs children. For example, the development of social bots for improving the children's social skills [1], chatbots for improving conversation abilities [2], and virtual reality systems for improving the children's ability to understand social emotional situations [3].

In the core of a system that can track and alert when any harmful situation occurs must be a crucial component that can detect these situations. Since some abusive or dangerous situations can be better detected through audio-based input sources, such as shouting or loud quarrels, while others are easier to be detected through text, such as a

child being persuaded to enter a stranger's vehicle, we believe that the core component should receive input as both text and audio. Indeed, combining multiple sources of information can provide a more comprehensive understanding of the situation, enabling the automated system to make more accurate conclusions and diagnose risks or unpleasant conditions more precisely. Therefore, in this paper we utilize machine learning (ML) and deep learning (DL) methods for detecting insulting or unsafe situations through text and audio of speech. To that end, we collected a dataset that includes the text and audio of over 1891 Hebrew sentences extracted from videos presenting real-world situations. Sentences were categorized into three classes: neutral sentences, insulting sentences, and sentences indicating unsafe conditions.

We used ML and DL methods for detecting unsafe and insulting conditions using the text and audio content of these sentences. We compared the performance of different ML methods, including a method that applies DL to text embedding using bidirectional encoder representations from transformers (BERT) [4,5] and audio embedding using Wav2Vec [6] for detecting unsafe and insulting situations. BERT has a clear advantage over traditional methods due to its ability to capture context, dependencies between words, and nuanced language. Additionally, fine-tuning BERT on specific datasets can improve accuracy on NLP tasks by allowing the model to learn the characteristics of the data and improve its performance on that task.

We found that the information extracted from the spoken text is more important for detecting unsafe and insulting sentences than that from the audio only, and that using DL applied to BERT fine-tuned sentences improved the accuracy of the results. However, audio signals have some added value, that is, a system trained on both text and audio signals achieved a slightly higher accuracy level than a system trained only on text.

Our results illustrate the technological ability to build an automatic agent that will be able to detect unsafe and unpleasant situations that children with special needs may encounter using the context of dialogues conducted with these children. This agent may first convert the audio input to text, and then use text embedding, in addition to wav embedding, as input for a neural network to determine unsafe situations. The ability to successfully detect unsafe and insulting contexts using embedded text and audio indicates the applicability of building such systems to assist and protect children with special needs. While various classical ML methods as well as DL based models have been developed for sentiment analysis and hate speech detection [7–10], to the best of our knowledge, no previous work has considered detecting harmful real-life situations, for children with special needs, based on multimodal input.

The remainder of this paper is organized as follows. Section 2 reviews studies that have already been conducted in the field of analyzing text and audio content and on conversational agents supporting children with special needs. Section 3 presents the composition of the dataset, and Section 4 presents the various methods used to obtain our results. Section 5 presents the results of various experiments. Finally, Section 6 provides conclusions and raises several open-ended questions that are relevant for further consideration in this study.

## 2. Related Work

In the following section, we have detailed several related works concerning software to assist children with special needs and state-of-the-art studies on text and audio emotion recognition.

### 2.1. AI and ML Application for Children with Special Needs

Many disabilities fall under the category of special needs, including autism spectrum disorder (ASD), down syndrome, Asperger syndrome, and Rett syndrome, all of which affect both children and adults. ASD is a lifelong neurodevelopmental disorder characterized by impaired reciprocal social communication and a pattern of restricted, often non-adaptive, repetitive behaviors, interests, and activities [11]. One widely accepted cognitive explana-

tion for these symptoms in people with ASD is the theory of mind (ToM). ToM refers to the ability of individuals of imputing mental states, such as emotions, beliefs, and ideas, to oneself and others, predicting the behavior of others on the basis of their mental states, and understanding social cues [12–14]. Difficulties in performing tasks related to ToM can impair social interaction, including deficits in pragmatic abilities and empathy [15]. These deficits might lead a person to make insulting statements unwittingly or be unaware of verbal bullying. A high prevalence of verbal bullying in children and adults with ASD has been documented, such as name calling and teasing (summarized in [16]).

In recent years, researchers have expressed a growing interest in using conversational agents and social robots as positive interventions for children with special needs [2]. In particular, several studies have shown that social robots can help improve social skills of children with ASD [17], and some have indicated that a child with ASD might find it easier to interact with a social robot than with a human teacher [18].

Xuan et al. [2] developed a chatbot dedicated to children with ASD to improve their conversation abilities. The chatbot was intended to arouse curiosity in children and improve their understanding of the conversation and used a large question-and-answer corpus.

Boucenna et al. [18] provided a comprehensive review of technologies, algorithms, interfaces, and sensors that can sense the children's behavior, train and improve their social abilities, and train individuals to recognize facial emotions, emotional gestures, and emotional situations. They suggested the use of robots to provide feedback and encouragement during skill-learning interventions and emphasized that a child with ASD might find it easier to interact with a robot than with a human teacher. The robot can provide instructions to the child and encourage them.

Recent research on technology-facilitated diagnosis and treatment of children and adults with ASD has been reviewed by Liu et al. [19]. They focused on the engineering perspective of autism studies and outlined three major delivery types of technology-facilitated autism studies: (1) Computers, game consoles, and mobile devices. (2) Virtual reality systems and devices. (3) Social robots. The first category included the use of serious games, which can be used to train an individual on many different skills in diverse contexts and situations, some of which may simulate real life.

In our ongoing research, we developed a social agent that should accompany children with special needs in their daily social interactions. Our study differs from the aforementioned studies; in our study, the agent must put itself in the place of the child, and assist him/her in the real world; in the studies described above, the role of the social agent is to be a friend or teacher of the child, and train the child's skills.

## *2.2. Technologies behind Emotion Recognition*

To help a child with special needs understand his/her environment and improve his/her social skills, we developed text and audio classifications to distinguish between different ongoing situations that the child may encounter. In this section, we discuss several studies focusing on emotion recognition via text and voice.

### 2.2.1. Text Emotion Recognition

Emotion recognition or sentiment analysis of text and speech is often used to determine the sentiments and emotions of writers or speakers [20]. Acheampong et al. [7] surveyed models, concepts, and approaches for text-based emotion detection (ED), and listed the important datasets available for text-based ED. In addition, they discussed recent ED studies and their results and limitations. A wide range of algorithms, including both supervised and unsupervised methods, have been employed for text-sentiment analysis. Early studies used several types of supervised ML methods (such as support vector machines (SVM), maximum entropy, and naïve Bayes (NB)) and a variety of feature combinations. Unsupervised methods include methods that exploit sentiment lexicons, grammatical analysis, and syntactic patterns. DL has emerged as a powerful ML technique and has achieved impressive achievements in many application domains, including sentiment analysis [8].

Another approach was used by Shaheen et al. [9], who proposed a framework for emotion classification in sentences, in which emotions were treated as generalized concepts extracted from sentences. They built an emotion seed called an emotion recognition rule and used k-nearest neighbors (KNN) and point mutual information classifiers to compare the generated emotion seed with a set of reference emotion recognition rules. In our previous work [21], we concentrated on insulting sentence recognition using only text content. We generated a dataset consisting of insulting and non-insulting sentences and compared the ability of different classical ML methods of detecting insulting content.

We have further described studies that apply DL methods to sentiment analysis. Socher et al. [22] proposed a semi-supervised recursive autoencoder network for sentence-level sentiment classification, which obtains a reduced dimensional vector representation of a sentence. In addition, Socher et al. [23] proposed a matrix-vector recursive neural network, which builds representations of multiword units from single-word vector representations to form a linear combination of the single-word representation.

Kalchbrenner et al. [24] proposed a dynamic convolutional neural network (CNN) using dynamic k-max pooling for the semantic modeling of sentences. Dos-Santos and Gatti [25] used two convolutional layers to extract relevant features from words and sentences of any size to perform sentiment analysis of short texts. Guan et al.'s [26] goal was to identify semantic orientation of each sentence (e.g., positive or negative) in a review. They proposed a weakly supervised CNN for both sentence- and aspect-level sentiment classifications. In our first stages of research, we combined several DL and context-sensitive lexicon-based methods.

Recent studies [27–32] applied a long short-term memory (LSTM) network for sentiment analysis. Akhtar et al. [33] employed several ensemble models by combining DL with classical feature-based models for the fine-grained sentiment classification of financial microblogs and news. This approach achieved slightly better results than Guggilla [30], who used LSTM for sentiment analysis on a dataset with similar properties. Wang et al. [29] proposed a regional CNN-LSTM model that consists of two parts: a regional CNN and an LSTM network for predicting the valence arousal ratings of text. However, the results obtained using this method were not as strong as those obtained using previous methods.

Our work is different from typical sentiment analysis, in which the emotions of the writer are detected; instead, we focused on detecting sentences that cause the listener to feel insulted or bullied. This will allow us to guide the child toward more appropriate behavior in the future; for example, choosing not to tell grandma that she is fat. A similar goal was addressed by Kai et al. [34]. In their study, Kai et al. used a rule-based system with underlying conditions that trigger emotions, based on an emotional model. The dataset comprised text from Chinese microblogs. They used an emotion model to extract the cause components of fine-grained emotions. Gui et al. [35] addressed the issue of emotion cause extraction, extracting stimuli, or the cause of an emotion. They proposed an event-driven emotion cause extraction method, in which a seven-tuple representation of events was used. Based on this structured representation of events and inclusion of lexical features, they designed a convolutional kernel-based learning method to identify emotion-causing events using syntactic structures.

Chkroun and Azaria [36,37] developed Safebot, a chatbot system that converses with humans. Safebots use human feedback to identify offensive behaviors. When Safebot was told that it said something offensive, it apologized and added the offensive sentence to its database. It then avoided using those sentences. While our proposed artificial assisting agent generally relies on its ability to perform sentiment analysis, it also relies on its ability of detecting hate speech, bullying, and insulting speech from two perspectives. From the perspective of children, we would like to detect bullying directed at children with special needs to protect them. From the viewpoint of those who interact with these children, we would like to observe that the child behaves appropriately and refrains from speaking in an insulting manner. The insulting sentences in our domain can be a result of

innocent intentions, and in most cases, they do not contain language that is considered as a bullying behavior.

Some prior work addressed the aforementioned issue. Nobata et al. [38] used Vowpal Wabbit's regression model and natural language processing (NLP) features to detect hate speech in online user comments from two domains. This approach outperformed a state-of-the-art DL approach. Libeskind et al. [39] detected abusive Hebrew texts in comments on Facebook by using a highly sparse n-gram representation of letters. Since comments on social media are usually short, they suggested four dimension reduction methods that classify similar words into groups and showed that the character n-gram representations outperformed all the other representations. Dadvar et al. [40] proposed integrating expert knowledge into a system for cyberbullying detection. Using a multicriteria evaluation system, they obtained a better understanding of bullying behavior of YouTube users and their characteristics through expert knowledge. Based on this knowledge, the system assigned a score to users that represented their level of bullying based on the history of their activities.

In a related study, Schlesinger et al. [41] focused on race talk and hate speech. They described technologies, theories, and experiences that enabled the conversational agent to handle race talk and examine the generative connections between race, technology, conversation, and CAs. By drawing together the technological-social interactions involved in race talk and hate speech, they pointed out the need for developing generative solutions focusing on this issue.

In our study, we utilized Heb-BERT [5], a Hebrew version of BERT, to analyze and classify texts. The BERT model's ability to capture context and meaning is useful for understanding and analyzing natural language data. Additionally, its fine-tuning capabilities allow for high accuracy in analyzing specific datasets. By utilizing BERT for research, we gain a more nuanced and accurate understanding of natural language data, making it possible to identify patterns and trends that are challenging to detect using other methods.

To evaluate the performance of Heb-BERT in our study, we compared the accuracy of various machine learning and deep learning methods on TF-IDF sentences, as well as on the Heb-BERT version of the training set, and we also fine-tuned the model using our training data. We achieved notably higher accuracy in detecting abusive and harmful sentences on the Heb-BERT representation of the test set compared to the results obtained by learning methods performed on the TF-IDF representation of the test set. This highlights the significance of using advanced natural language processing techniques such as Heb-BERT in research, especially in situations where accurate detection is critical.

### 2.2.2. Voice Emotion Recognition

Speech is the fastest and most natural mode of communication among humans. This has motivated researchers to consider speech as an efficient method for human–machine interaction.

El Ayadi et al. [42] survey three important aspects of the design of a speech emotion recognition system: the choice of suitable features for speech representation, the design of an appropriate classification scheme, and the proper preparation of an emotional speech database for evaluating system performance. Feature selection is a crucial step in the process of audio based sentiment analysis. It involves identifying the most relevant features or characteristics of an audio signal that can be used to effectively predict the sentiment expressed in it. This can greatly improve the accuracy and efficiency of sentiment analysis algorithms, as well as reduce the amount of data that needs to be processed.

There are several methods for feature selection in audio-based sentiment analysis, each with its own strengths and weaknesses. Some of the most common methods include:

- Mel-frequency cepstral coefficients (MFCCs) capture the spectral envelope of the audio signal. They can capture the prosodic and emotional cues in speech, such as pitch and intonation Nwe.

- Prosodic features [43] refer to the rhythmic, dynamic, and intonational aspects of speech, such as speaking rate, emphasis, and pitch. These features are closely tied to the emotions expressed in speech.
- Spectral features [44] refer to the distribution of energy across different frequency bands in the audio signal. They can capture the acoustic features of speech, such as the formants and harmonics.

Soleymani et al. [45] review sentiment analysis studies based on multimodal signals, including visual, audio and textual information. The source data comes from different domains, including spoken reviews, images, video blogs, human–machine and human–human interactions. In our study, the data source used for the risk or insulting context detection, was based on human–human interactions, obtained from YouTube videos of Hebrew children's movies, and the speech text was extracted from the recordings manually by research assistants.

Vocal speech can be classified using classical ML methods. Noroozi et al. [46] proposed the use of random forest for vocal emotion recognition. This technique adopts random forests to represent speech signals along with the decision-tree approach to classify them into different categories. The emotions were broadly categorized into six groups. The Surrey audio visual-expressed emotion database [47] was used in this study. The proposed method achieved an average recognition rate of 66.28%.

Han et al. [48] proposed using a deep neural network (DNN) for producing an emotion state probability distribution for each speech segment. Nwe et al. [49] proposed a method representing speech signals and discrete hidden Markov model (HMM) as a classifier using short-time log frequency power coefficients (LFPC). Performance of the LFPC feature parameters was compared with that of the linear prediction Cepstral coefficients (LPCC) and mel-frequency Cepstral coefficients (MFCC) feature parameters commonly used in speech recognition systems. Results show that the proposed system yielded an average accuracy of 78% Li et al. [50] recently combined DNN and HMM using acoustic models that achieved good speech recognition results over Gaussian mixture model-based HMMs, with a restricted Boltzmann machine (RBM), and achieved an accuracy of up to 77.92%.

A different type of neural network was used by Wu et al. [51]. They explored spectrogram-based representations for speech emotion classification using the USC-IEMOCAP dataset. They experimented with features from both the speech and glottal volume velocity spectrograms. Their experiments investigated whether classification performance could be improved by filtering out unwanted factors of variation, such as speaker identity and verbal content (phonemes), from speech.

Zadeh et al. [52] introduced a tensor fusion Network (TFN) that learns both intramodal and intermodal dynamics end to end. The intermodal dynamic was shaped using a fusion approach called Tensor Fusion, which explicitly accumulates unimodel, bimodal, and trimodal interactions. The intermodal dynamics were modeled through three sub-networks that embed models for languages, visual, and acoustic, respectively, and achieved 69.4% accuracy for binary classification.

Jain et al. [53] used a SVM to classify speech as one of four emotions (sadness, anger, fear, and happiness). They classified these emotional states using a SVM classifier using two strategies: one against all (OAA) and gender-dependent classification. They achieved 85.085% accuracy when using MFCC data.

Our final goal was online emotion recognition to mediate the environment for children with special needs. Research has been conducted on developing online speech emotion recognition (SER) systems. Bertero et al. [54] built a conventional dialogue system based on modules that enable them to have "empathy" and answer the user while being aware of their emotions and intent. They used a CNN model to extract emotions from raw speech inputs, without feature engineering. This approach achieved accuracy of 65.7%.

Gandhi et al. [10] survey recent developments in multimodal sentiment analysis, including several fusion technologies, popular multimodal datasets, deep learning models for multimodal sentiment analysis, interdisciplinary applications, and future research direc-

tions. Another application of multimodal input of text and audio is sarcasm detection [55]. Castro et al. created a dataset that includes highly-annotated conversational and multi-modal context features. The dataset also provides prior turns in the dialogue, which serve as contextual information.

Another interesting work on text- and voice-based categorization was performed by Zhu et al. [56]; they suggested combining BERT text-embedded vectors with Wav2Vec audio-embedded vectors for dementia detection. They used the Wav2Vec model to generate automatic speech recognition (ASR) transcripts and vectors to fine-tune BERT, followed by inference layers consisting of a convolutional, global average pooling, and fully connected layers for the dementia detection task. Dementia detection was performed based on the BERT embedding vector, where pauses in the audio were recorded as punctuation marks. While speech slowing down and pauses in speech can be a sign for dementia, in our problem, there are almost no clear audio features. Although some audio features do exist, such as a specific type of intonation expressing anger, they do not have clear textual features that can be converted to BERT as performed by Zhu et al. Thus, for the classification task we utilized both Wav2Vec and Hebrew BERT embedded vectors to detect risky and insulting sentences. In particular, we used the Wav2Vec model for fine-tuning and extracting audio features from our data, combined the given embedded vector with the text Heb-BERT embedded vector [5], and used a concatenate vector to train our DNN model. We achieved accuracy of up to 80%.

## 3. Dataset Details

In this section, we describe the process of creating a corpus of Hebrew-tagged sentences using their soundtrack files. This process consists of the following steps:

### 3.1. Audio Tracks

In the first step, we considered several sources of open videos, obtained from YouTube (https://www.youtube.com/, accessed on 15 March 2023) videos of Hebrew movies for children with child actors. We obtained various YouTube videos from different sites that dealt with teaching complex social-situation-based relevant behavior, and from parents who shared different recordings and news sites documenting different dialogues between people. In particular, the source films were obtained from several known Israeli-film series, which were stored on YouTube in the public domain. The films were obtained from the "Chaim Neighborhood", "Asi and Tuvia", and "Between the Rings" series. In addition, we downloaded several other YouTube films, including videos that were filmed and produced independently by children in Israel and uploaded on YouTube, and videos filmed and produced by various Israeli high schools as part of film production programs. For each movie, we downloaded the audio, chose contexts regarding conversations between children in various social situations, and split the conversations into sentences. Each voice sentence was separately stored as an audio mp3 file, whereas the sentence texts were collected in one csv file. We then proceeded with tagging, as described in the next section. The resulting dataset included a total number of 3975 Hebrew sentences, where each sentence consisted of its own text and audio.

### 3.2. Sentences Tagging

For tagging, we employed three catalogers from the field of education. The first is a special education student, the second is a preschool student, and the third has a B.Ed. degree in education. Each cataloger received his/her own list of sentences and soundtracks, and depending on the text and audio, categorized each sentence in one of the following categories: (a) normal sentences, (b) insulting sentences, and (c) unsafe sentences, which may indicate danger to the child. Insulting sentences were classified into two types: (b1) sentences insulting a second person and (b2) sentences insulting a third person. However, to have enough samples of each type, we combined the two types into one, as explained in Section 3.3. In addition, there were two other categories: (d) sentences that could offend

if they were said in a certain context, and finally, (e) sentences expressing an excess of affection, which could indicate dangerous situations for the child if said by strangers.

Note that: "insulting speech" refers mostly to unpleasant or tactless sentences, such as: "Grandma, your food came out burnt today!", while "unsafe speech" means sentences that indicate a risky situation for the child, such as: "Let's run to the road!". The difference between these two categories is important for an assistance agent, who should indicate insulting situations in which the child with ASD talked in a way that can be impolite, while in risky situations, the assistance agent should alarm the child educators and/or parents about the risky situation he/she was involved in.

Each sentence was tagged with the help of the three catalogers. When three or two of them agreed on one category, that category was selected as the sentence tag. However, when each cataloger categorized the sentence differently, the sentence was removed from the study. After categorizing the sentences, we obtained the following lists of tagged sentences: neutral speech, with no risk and unpleasantness (1507 sentences), and sentences that may be insulting, depending on the context (326 sentences), second-person insulting speech (368 sentences), third-person insulting speech (351 sentences), unsafe speech (473 sentences), and sentences expressing affection (53 sentences). An additional 897 sentences were classified differently for each classifier.

### 3.3. Dataset Balancing

Because the number of insulting sentences in each sub-category (second- and third-person) was relatively small, we merged both sub-categories into a single category, called insulting sentences. In addition, because of the small number of context-dependent and affection sentences, we did not include them in the current study. Finally, we observed that the number of neutral sentences was significantly higher than that of sentences in the other two categories (insulting and unsafe). Thus, we chose 700 neutral sentences (360 of them were labeled as neutral by all classifiers and the rest were randomly chosen). In addition, sentences of different grades by each classifier were not included in the dataset.

### 3.4. The Final Dataset

After the aforementioned process, we obtained a dataset composed of 1891 Hebrew sentences, including both text and audio. The final sentence distributions are presented in Table 1.

**Table 1.** Category example and distribution.

| Category Name | Example | Total Number |
|---|---|---|
| Neutral Sentences | How are you? | 700 |
| Insulting (1st and 2nd person) sentences | She is fat | 719 |
| Unsafe (risky) sentences | It's dangerous! | 472 |

Figure 1 graphically shows the distribution of the three categories in the dataset. All software and data are available upon request. As shown in Table 1 and Figure 1, we used a balanced number of sentences for each of the above three categories.

### 3.5. Dataset Analysis

Next, we analyzed the common words for each category in the dataset. Figures 2–4 presents the frequency of the 20 most common words for each category, while omitting stop words that were often common.

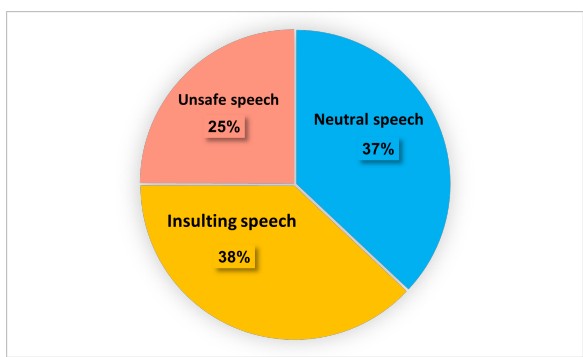

**Figure 1.** Dataset distribution into categories.

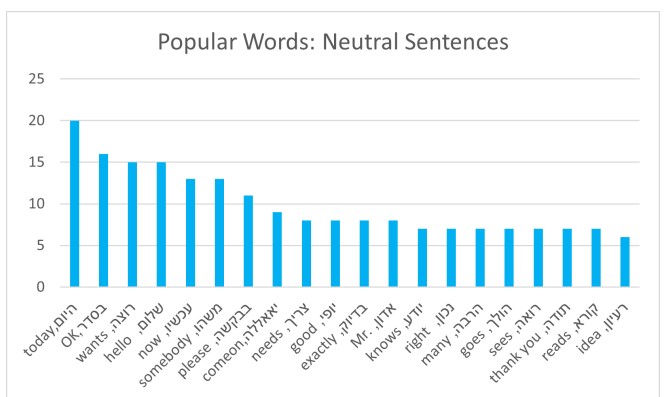

**Figure 2.** Frequency of neutral words—no stop words.

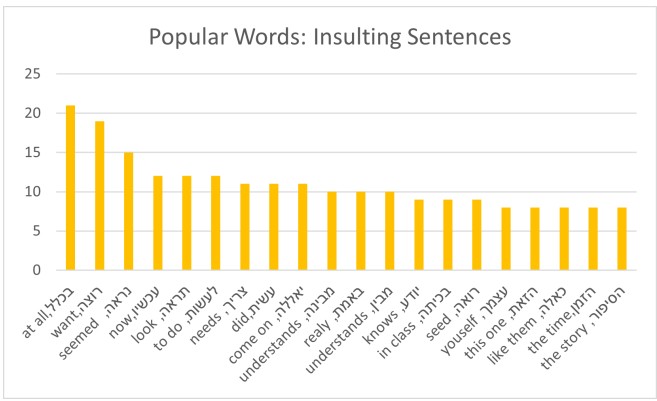

**Figure 3.** Frequency of insulting words—no stop words.

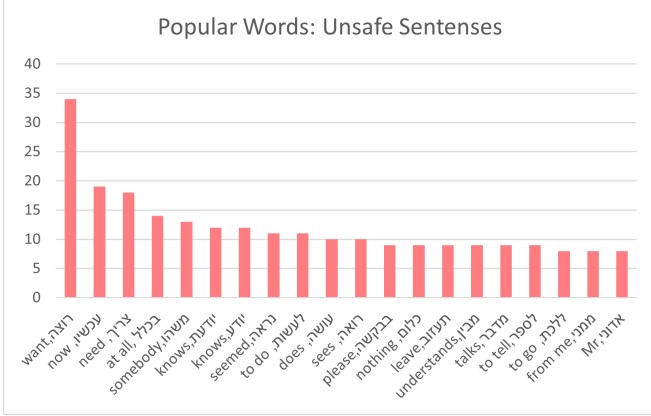

**Figure 4.** Frequency of unsafe words—no stop words.

As can be seen, each category exhibited its own set of popular terms and words, with some overlap between common terms. Figure 5 shows the distribution of the common terms.

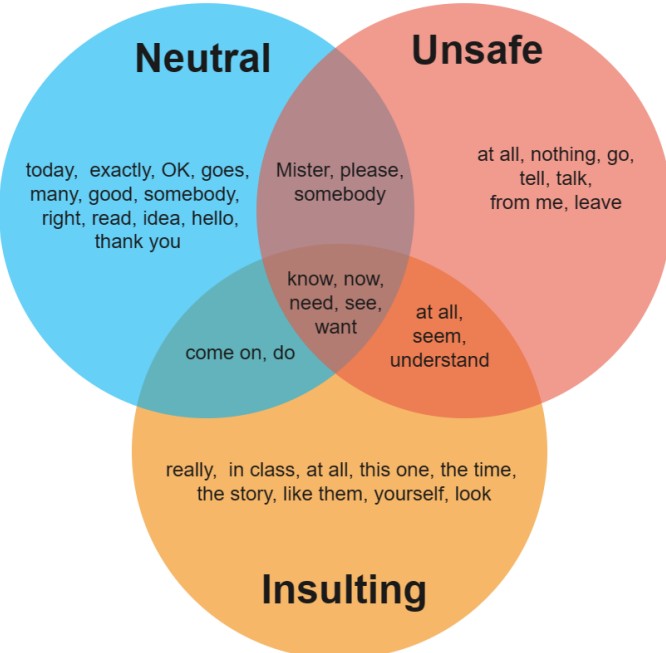

**Figure 5.** Common terms for each Category.

This figure demonstrates the uniqueness of each category with respect to its own popular terms, with some overlap between terms that are popular in two or even three categories. However, identifying unique text characteristics for each category, which will allow us to accurately classify sentences based on their texts, is possible. Next, we describe the five audio properties of the dataset [57].

1.  MFCC [58] is a set of fundamental audio features, using a 20 ms audio frame unit. It also includes the noise, speech rate, speech acceleration etc.
2.  Mel-scale spectrogram converts the frequencies to the mel scale, which is similar to human ear as they are equidistant from one another.
3.  Spectral contrast [44] is the difference in amplitude between the spectral peaks and valleys for six sub-bands for each time frame. It can be used to highlight regions of the frequency spectrum.
4.  Short-time Fourier transform (STFT) [59] takes place around a short time and evaluates the Fourier return on the time-dependent segment. The Chroma feature relates to the twelve different pitch classes and provides a robust way to describe a similarity measure between audio pieces.
5.  Tonnetz is a tonal space representation introduced by Euler [60] and is used for detecting Harmonic Change in Musical Audio [61].

Table 2 and Figure 6 show the average values of each characteristic for each category.

**Table 2.** Average value (and standard deviation) of each parameters for each category.

| Feature | Neutral | Insulting | Unsafe |
| --- | --- | --- | --- |
| MFCC | −5.76 (2.344) | −5.07 (2.9) | −5.07 (2.75) |
| Mel | 1.45 (2.12) | 1.612 (2.41) | 1.91 (2.72) |
| STFT Chroma | 0.55 (0.07) | 0.556 (0.07) | 0.56 (0.08) |
| Contrast | 22.021 (1.95) | 21.83 (1.73) | 21.71 (1.82) |
| Tonnetz | 0.0078 (0.03) | 0.0067 (0.03) | 0.0093 (0.02) |

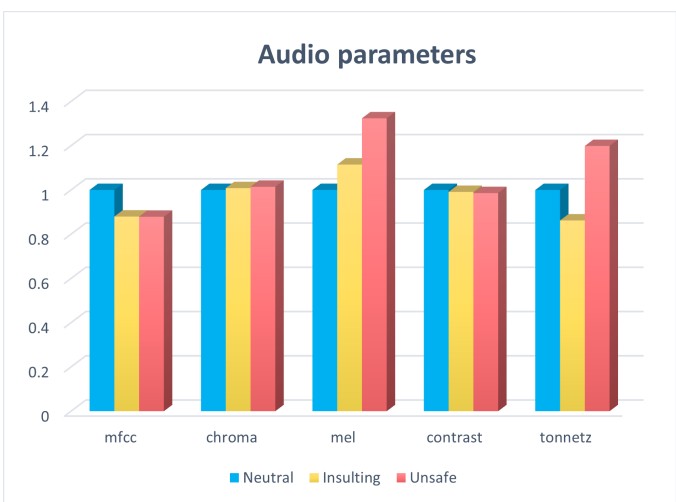

**Figure 6.** Average value of each audio parameter, divided by the neutral average value.

As can be seen in the table, both the Tonnetz and Mel-scale spectrograms have the highest value in unsafe speech. This indicates that, in unsafe speech, more variety existed in the audio and exhibited a more varied tone. MFCC exhibited the lowest value in neutral speech, probably because it is related to the normal human ear. The contrast exhibited the highest average value for neutral sentences. The average value of the Chrome feature was very similar among the three classes.

Finally, we attempted to extract a vector of the eight popular emotions based on audio signs. As classified by the Wav2Vec2 Feature Extractor (https://huggingface.co/ehcalabres/wav2vec2-lg-xlsr-en-speech-emotion-recognition, accessed on 15 March 2023). Table 3 shows the emotion level specified by each categories, for each of the following emotions: ''angry'', ''calm'', ''disgust'', ''fearful'', ''happy'', ''neutral'', ''sad'', and ''surprised''. Some of the values obtained were reasonable; for example, insulting sentences express more anger and more disgusting emotions than neutral and unsafe sentences, but other values, such as higher values of happiness and lower values of fear for unsafe sentences, were more difficult to explain. This might be one of the reasons of poor performance of predicting sentence categorization given these eight extracted emotions, as described in Section 5 below.

**Table 3.** Emotions for each category.

| Emotion Type | Neutral | Insulting | Unsafe |
|---|---|---|---|
| Angry | −0.0045 | −0.0003 | −0.0022 |
| Calm | −0.0054 | −0.0070 | 0.0065 |
| Disgust | −0.0039 | 0.0001 | −0.0058 |
| Fearful | −0.0021 | −0.0035 | −0.0091 |
| Happy | 0.0011 | 0.0014 | 0.0060 |
| Neutral | 0.0001 | 0.0008 | 0.0048 |
| Sad | −0.0033 | −0.0040 | −0.0051 |
| Surprised | 0.0048 | 0.0008 | −0.0008 |

## 4. Methodology Description

As mentioned above, we intended to develop an agent that would help children with special needs to understand the environment and behave as expected in a variety of situations. Therefore, we tested whether the addition of audio improved the performance of our model. To detect insulting and harmful content given by text and voice, we used text recognition methods and methods that combined text and audio recognition. The exact hyperparameters used for each method can be found in the project implementation (https://github.com/MLspecialNeeds/harmful_sentence_detection, accessed on 15 March 2023).

*4.1. Methods Used for Classification via Text Features*

We started with the database of Hebrew text sentences and applied different ML methods to the sentences. The text used as input for the different ML methods is given in the following formats:

1. **TF-IDF** vectors: [62] generated by Sklearn Tfidf Vectorizer (https://scikit-learn.org/stable/modules/generated/sklearn.feature_extraction.text.TfidfVectorizer.html, accessed on 15 March 2023).
2. **Heb-BERT** format, as explained below.
3. Fine-Tuned Heb-BERT: Heb-BERT vectors, fine-tuned on our dataset, as explained below.

We further explain the Hebert and fine-tuned Heb-BERT formats used for text vectorization.

**BERT** (bidirectional encoder representations from transformers) is a transformer-based ML technique developed for natural language processing (NLP) pretraining. A transformer is an attentional mechanism that learns relationships between words (or subwords) in a text. It has two separate mechanisms: an encoder that reads the text input and a decoder that produces a prediction of the task. To improve the classification accuracy, we used pretrained embedding models for text inputs. Because our dataset was composed of Hebrew sentences, we used the pre-trained **Heb-BERT** model [5] to transform the sentences into embedded structures. Heb-BERT is based on the BERT architecture [4]. Similar to BERT, it is used for diverse NLP tasks, especially for sentiment analysis. As described in Section 5.1, using classical ML methods on embedded inputs improves the accuracy of the classification. Finally, we ran the BERT fine-tuning method on the Heb-BERT model for sentiment analysis, as described by Chris Tran (https://skimai.com/fine-tuning-bert-for-sentiment-analysis/, accessed on 15 March 2023).

These three text formats, namely, term frequency–inverse document frequency ( TF-IDF), Heb-BERT, and fine-tuned Heb-BERT transformers, were applied to the Hebrew dataset, and the resulting vectors were used as inputs to the ML methods. Our results, presented in Section 5, show that the fine-tuned Heb-BERT transformer achieved the highest accuracy in detecting insulting and unsafe situations. Thus, when combining text and audio information, we use BERT and fine-tuned BERT combined with the vectorized representation of the audio, as explained in Section 4.3.

We considered the following classical ML methods for classification based on text content:

1. **MLP**: A fully connected neural network with the following layers: the first hidden layer size is identical to the input size, followed by a hidden layer of size 100 and 50. The tanh activation function was used in all hidden layers. The final output layer consisted of a sigmoid neuron for binary classification, and three softmax neurons for the task of trinary classification (neutral, insulting, or unsafe sentenses). The loss function was the weighted cross-entropy, and training was performed using the ADAM optimizer with a learning rate of 0.0001.
2. **SVM**: A set of supervised learning methods used for classification (see Section 4.2 for additional details).
3. **KNeighbors**: An unsupervised learning method based on a similarity measure between data-points (see Section 4.2 for additional details).
4. **Random Forest**: An ensemble learning method for classification (see Section 4.2 for additional details).
5. **ExtraTrees**: A method that combines the predictions of several decision trees.
6. **Logistic regression**: A model that is used when the value of the target variable is categorical in nature. The softmax algorithm is used for more than two categories.
7. **NB**: A classification technique assuming the predictors to be independent based on Bayes' theorem (see Section 4.2 for additional details).

8. **Voting**: A combination of all the above ML classifiers. This method uses a majority vote or the average predicted probabilities to predict the labels (see Section 4.2 for additional details).

*4.2. Methods Used for Classification via Audio*

In this section, we first describe the details of the models used for audio representation and then provide the ML methods used for the classification tasks using audio features only.

4.2.1. Audio Representation

The following formats were used to represent audio data:

1. **Eight emotions vector** This eight popular emotions vector, as classified by the Wav2Vec2 Feature Extractor (https://huggingface.co/ehcalabres/wav2vec2-lg-xlsr-en-speech-emotion-recognition, accessed on 15 March 2023).
2. **FSFM vector** The five sound feature model (FSFM) vector is constructed by concatenating the five audio features described in Section 3.5, namely, MFCC, mel-scale spectrogram, spectral contrast, STFT, and Tonnetz. Concatenation of the above feature vector resulted in a vector with length of 193, and this vector was provided as an input to the multilayer perceptron (MLP) network [57].
3. **MFCC subframed vectors**: MFCC, 15 audio frames units of 20 ms [58] obtained from the audio samples, were used as inputs to the bidirectional recurrent neural network (RNN) model.
4. **Wav2Vec-BASE Pretrained vectors** (https://huggingface.co/facebook/wav2vec2-base, accessed on 15 March 2023): a speech model that accepts a float array corresponding to the raw waveform of the speech signal. Wav2Vec2 model was trained using connectionist temporal classification (CTC), hence the model output, a float vector with length of 768, was decoded using Wav2Vec2CTCTokenizer.
5. **Fine-tuned Wav2Vec-BASE vectors**: BASE vector was fine-tuned on our dataset.
6. **Wav2Vec for Emotions**: an embedded vector with length of 1024 was fine-tuned for emotion classification.
7. **Fine-tuned Wav2Vec for Emotions**: emotions embedded vector was fine-tuned on our dataset.

4.2.2. Wav2Vec 2.0 Pretrained Models

The Wav2Vec 2.0 pretrained models, developed by Facebook AI. [6], is a framework for self-supervised learning of vector representation from speech audio, pretrained in large quantities of audio data. This model attempts to recover a randomly masked portion of the encoded audio feature. The model consisted of three main modules. The first module was a feature encoder, composed of a 1-CNN encoder that downsamples the input raw waveform $\mathcal{X}$ to a latent speech representation of 25 ms each $\mathcal{Z}$ in T time steps. The second module was a contextualized encoder consisting of several transformer encoder blocks that transform latent representations $\mathcal{Z}$ into contextualized representations $\mathcal{C}$. In addition, a quantization module discretized the speech representation $\mathcal{Z}$ into a finite set of quantized representations $\mathcal{Q}$ by matching it with a code-book for selecting the most appropriate representation of the audio. Its objective was to identify these quantized representations of the masked features using the output of the contextualized network $\mathcal{C}$ for each masked time step $T$ using the contrastive loss function. After pre-training the unlabeled audio data, the model can be fine-tuned on the labeled data to be used for downstream tasks.

In this study, we compared the accuracy of insulting and unsafe situation detection using text and audio content of spoken sentences in different situations. In particular, for detection through audio, we used both the Wav2Vec 2.0, base model, called *Wav2vec Base* (https://huggingface.co/facebook/wav2vec2-base, accessed on 15 March 2023, and a model that was fine-tuned on the SER task, called *Wav2Vec for Emotions*. Their vector embedding sizes were 768 and 1024, respectively. In addition, we fine-tuned both models using our dataset.

The Wav2vec 2.0 Emotion pretrained model (https://huggingface.co/ehcalabres/wav2vec2-lg-xlsr-en-speech-emotion-recognition, accessed on 15 March 2023) was a fine-tuned model based on the Wav2Vec 2.0 xlsr-53 model (https://huggingface.co/jonatasgrosman/wav2vec2-large-xlsr-53-english, accessed on 15 March 2023), which was trained using the audio dataset [63] currently consisting of 7335 hours of transcribed speech in 60 languages. The Wav2vec 2.0 emotion pretrained model used in this study is a model based on the Wav2Vec 2.0 model and was fine-tuned for emotion recognition on the RAVDESS dataset [64], a multimodal database of emotional speech and song that contains 1440 hours of samples in eight different emotion classes, recording professional actors, in English.

Wav2Vec 2.0 Fine-Tuning

In addition, we fine-tuned the Wav2Vec 2.0 base and Wav2Vec 2.0 emotion pretrained models using our dataset. During the fine-tuning process, we extracted the context representation of our data from the pre-trained models, starting with an average pooling layer calculating an averaged vector according to the time dimension, followed by a fully connected layer with the tanh activation function. Finally, a fully connected layer was used for the classification task. Because the Wav2Vec 2.0 model was used as a feature extractor, the weights of the feature encoder module of the pre-trained model were not changed during the fine-tuning process. This fine-tuning architecture was inspired by [65] because of its similarity to our task and its ability to achieve satisfactory results in their tasks. The hyper-parameters used for fine-tuning are listed in Table 4.

**Table 4.** Hyper-parameters used for fine-tuning.

| Parameter | Value |
| --- | --- |
| Sample frequency | 16 kHz |
| Learning rate | $2 \times 10^{-5}$ |
| Training epochs | 10 |
| Training batch size | 3 |
| Gradient accumulation steps | 2 |
| Total train batch size | 6 |

4.2.3. Machine Learning Classifiers

The audio formats described above were examined using the following ML classifiers:

1. **A fully connected neural network** with the first hidden layer identical to the input size, followed by a dropout layer of 0.3, a fully connected layer of size 100, an additional 0.3 dropout layer, and three additional fully connected layers of size 868, 100, and 50. Unless otherwise stated, the tanh activation function was used in all FC hidden layers. The model illustration is provided in Figure 7. The final output layer consisted of a sigmoid neuron for binary classification and three softmax neurons for the task of three types classification (neutral, insulting, or unsafe sentences). The loss function used was the weighted categorical cross entropy or binary-cross entropy, and training was performed using the ADAM optimizer with a learning rate of 0.0001.

2. **Bidirectional RNN classifier** An RNN model: This model was trained on the sub-framed MFCC, obtained from the audio samples. The model consisted of an LSTM cell with a ReLU activation function, followed by a dropout of 0.3, and a fully-connected layer with a sigmoid activation function (or softmax for three classes). It used a weighted cross-entropy loss function and ADAM optimizer with a learning rate of 0.0001.

Note that the NLP classifier was examined on the different audio vector formats, namely, the eight emotion vectors, FSFM vector, and different variants of the Wav2Vec vectors, while the bidirectional RNN classifier received the MFCC sub-frame data, presented as a sequence of audio signals over time, as inputs.

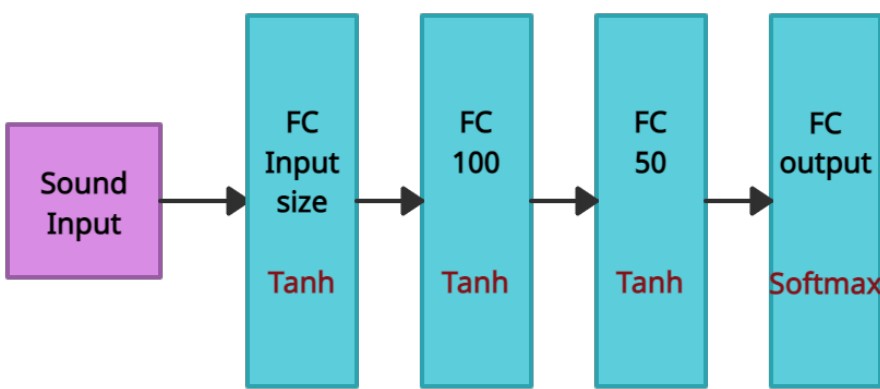

**Figure 7.** An illustration of the audio MLP model.

*4.3. ML Methods Used for Combined Text and Audio*

In the next set of experiments, we used both audio and textual features. We assumed the combination of audio and text for enhancing the results of the classification task. We applied different combinations of textual data and data extracted from the audio recordings.

To train both features, we built a *Wav2vec Emotion Vector* model that shrank the given audio Wav2Vec embedded vector to a size of 100, and we concatenated it with the text Heb-BERT embedded vector with size of 768, which was followed by the text MLP model. This model is shown in Figure 8. In our research on detecting insulting and harmful content, we discovered that the context of the text was more relevant in determining its sentiment than the tone in which it was delivered. Although a recording may depict a dangerous situation, it may still be spoken in a calm tone. To better account for the relevance of the text compared to that of the audio, we chose to reduce the size of audio inputs to a certain extent as they contribute less than text data. Nevertheless, it seems important to retain information from the audio as it still holds valuable sentiment analysis information.

The combined model was aimed at utilizing both textual and audio information to achieve greater accuracy in detecting challenging situations. First, we processed the text data using text Heb-BERT vectorization, as explained in Section 4.1. In addition, the audio vector, given in one of the formats described in Section 4.2.1 except the MFCC, was passed through a fully connected layer with 100 neurons and a ReLU activation function, with a 30% dropout layer before and after the fully connected layer. Then, both outputs (textual-based input and audio-based output) were merged, and three additional fully connected layers were used. This was followed by a softmax layer for the detection of unsafe/insulting/neutral situations and a sigmoid neuron for binary classification detection. This architecture provided the most accurate results, as described in the next section. For additional information, we refer the reader to the GitHub website (https://github.com/MLspecialNeeds/harmful_sentence_detection, accessed on 15 March 2023).

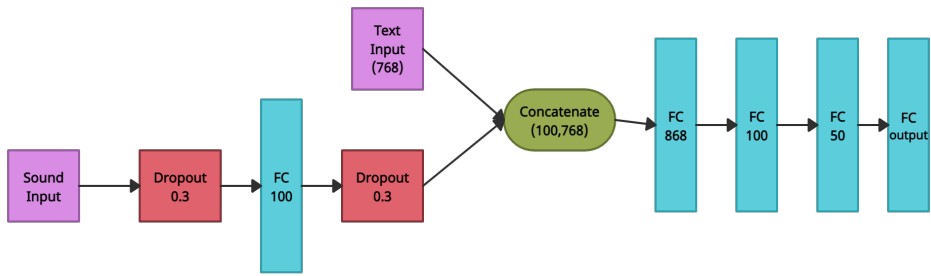

**Figure 8.** An illustration of the text and audio model.

Based on these various ML methods, in the next section, we compare the accuracy of the different models: text-based, audio-based, and combined text-and-audio-based models, in preset challenging situations.

## 5. Experimental Result

In the following section, we describe our experimental results of insulting and unsafe sentence content detection using text, audio, or both sources, and apply the ML methods described above.

In all our experiments, we tested all models using five-fold cross validation on our collected dataset, and the accuracy presented in the results table is the accuracy of the test set in all our experiments. In addition, for all experiments using the DNN, we ran a model with 100 epochs.

### 5.1. Results on Text-Based Inputs

First, we utilized classical ML methods for insulting and unsafe sentence detection using text sources. We ran each ML method over three different text sources: TF-IDF transformer applied to the sentences, Heb-BERT representation of the sentences, and Heb-BERT fine-tuned on our training set, where for each fold of the experiment, the Heb-BERT was trained on a different training sets (according to the fold), and then the result layer was used to create embedded vectors for all sentences. We used the weighted cross-entropy loss function in all ML methods, except for the KNN and voting methods. The voting method classified the examples using the majority rule of the classification of the following ML classifiers: random forests, extra trees, KNN, SVM, and ridge classifier. For the TF-IDF inputs, the NB classifier was also considered in the voting process. Our results are presented in Table 5 for insulting sentence detection, Table 6 for unsafe situation detection, and Table 7 for all three categories of classification.

**Table 5.** Accuracy of insulting sentences detection based on text only.

| Model | Tfidf | BERT | Fine-Tuned BERT |
|---|---|---|---|
| MLP | 0.72 | 0.83 | 0.86 |
| SVM | 0.75 | 0.84 | 0.86 |
| K Neighbors | 0.69 | 0.83 | 0.86 |
| Random Forest | 0.71 | 0.84 | 0.86 |
| Extra Trees | 0.74 | 0.84 | 0.87 |
| Logistic Regression | 0.75 | 0.83 | 0.86 |
| Naive Bayes | 0.75 | - | - |
| Voting | 0.75 | 0.84 | 0.86 |

**Table 6.** Accuracy of unsafe sentences detection based on text only.

| Model | Tfidf | BERT | Fine-Tuned BERT |
|---|---|---|---|
| MLP | 0.72 | 0.75 | 0.82 |
| SVM | 0.72 | 0.72 | 0.83 |
| K Neighbors | 0.66 | 0.69 | 0.83 |
| Random Forest | 0.72 | 0.73 | 0.83 |
| Extra Trees | 0.72 | 0.74 | 0.83 |
| Logistic Regression | 0.72 | 0.69 | 0.83 |
| Naive Bayes | 0.69 | - | - |
| Voting | 0.71 | 0.74 | 0.83 |

**Table 7.** Accuracy of classifiers for all three categories based on text only.

| Model | Tfidf | BERT | Fine-Tuned BERT |
|---|---|---|---|
| MLP | 0.56 | 0.57 | 0.72 |
| SVM | 0.61 | 0.55 | 0.75 |
| K Neighbors | 0.55 | 0.55 | 0.74 |
| Random Forest | 0.58 | 0.60 | 0.74 |
| Extra Trees | 0.60 | 0.59 | 0.74 |
| Logistic Regression | 0.60 | 0.55 | 0.75 |
| Naive Bayes | 0.58 | - | - |
| Voting | 0.62 | 0.58 | 0.74 |

Generally, using BERT inputs significantly improves the accuracy of insulting sentence detection for all ML classifiers, and running fine-tuning on BERT sentences results in an additional improvement. However, for unsafe sentence detection, the use of BERT sentences slightly improved the accuracy, while the use of fine-tuned BERT significantly improved the detection accuracy. For three-category classification, the use of BERT did not improve the accuracy, while the use of fine-tuned inputs significantly improved the accuracies of all ML methods.

In addition, different ML methods achieved different accuracies for TF-IDF inputs. The best results were obtained for the SVM and voting classifiers for the binary problems and three-category classification problems, respectively. However, the variance of the accuracy of different classifiers reduced, and nearly disappeared, when moving from TF-IDF inputs to Heb-BERT and fine-tuned Heb-BERT inputs, probably because most of the learning on the FT-based input was executed during the fine-tuning on the BERT examples.

Comparing other scores, a similar impact was observed when moving from TF-ITF inputs to BERT and fine-tuned BERT models. Table 8 shows the average F1 score for the SVM, MLP, and voting methods on the different text-based inputs (TF-IDF, BERT, and fine-tuned BERT), and for the different types of classification problems (insulting sentence, unsafe sentence, and three-category detections).

**Table 8.** F1 Score of classifiers for all three categories based on text only.

| Classification | Method | TFIDF | BERT | BERT-FT |
|---|---|---|---|---|
| Insulting | SVM | 0.75 | 0.85 | 0.86 |
| Insulting | MLP | 0.72 | 0.83 | 0.86 |
| Insulting | Voting | 0.75 | 0.84 | 0.86 |
| Unsafe | SVM | 0.70 | 0.71 | 0.82 |
| Unsafe | MLP | 0.70 | 0.72 | 0.82 |
| Unsafe | Voting | 0.68 | 0.70 | 0.82 |
| Three Categories | SVM | 0.59 | 0.54 | 0.74 |
| Three Categories | MLP | 0.56 | 0.50 | 0.67 |
| Three Categories | Voting | 0.58 | 0.50 | 0.73 |

We can conclude that application of ML to fine-tuned BERT vectors can dramatically improve the accuracy and other factors such as the F1 score and that the difference between the accuracy of the results obtained by different ML methods is reduced when using the fine-tuned BERT inputs. In addition, for the three-category classification, the use of Heb-BERT inputs reduces the average accuracy and average F1-score for most of the ML methods, but the use of fine-tuned BERT inputs increases the accuracy and F1 score for all the ML methods examined.

### 5.2. Results on Audio-Based Inputs

Our next set of experiments examined the ability of ML methods to correctly detect insulting and unsafe situations from audio files. We compared different audio representations as described in Section 4.2.1. The audio sources were derived from the recorded audio files, as described in section 4.2.1. In particular, the input of the RNN method consisted of MFCCs, whereas the other audio sources are analyzed by the DNN audio classifier described in Section 4.2.3. Recall from Section 4.2.1 that the following data were extracted for the audio-based prediction challenge: the Eight Emotions Vector, as classified by Wav2Vec2 Feature Extractor; FSFM Vector constructed from a concatenation of the five audio features, Wav2Vec-BASE Pretrained vectors with a length of 768, with or without fine-tuning on our data, and Wav2Vec embedding vectors, with length of 1024, fine-tuned for emotions, with or without fine-tuning on our data. Our results are presented in Tables 9–11. In fact, the accuracy achieved with audio-only prediction was much lower than that achieved with text-only prediction. This means that audio signals provide only partial clues about situations with respect to text information. We can also observe that the accuracy of situation predicting via eight emotions vector represented results close to those predicted via randomly created vectors. However, emotion prediction using fine-tuned vectors improved the prediction accuracy when only audio data were available.

**Table 9.** Accuracy of insulting speech detection based on audio only.

|                | Audio | FT Audio |
|----------------|-------|----------|
| RNN            | 0.60  | -        |
| Eight Emotions | 0.48  | -        |
| FSFM           | 0.60  | -        |
| Base           | 0.61  | 0.64     |
| Emotion        | 0.63  | 0.66     |
| Random         | 0.51  | -        |

**Table 10.** Accuracy of unsafe speech detection based on audio only.

|                | Audio | FT Audio |
|----------------|-------|----------|
| RNN            | 0.60  | -        |
| Eight Emotions | 0.50  | -        |
| FSFM           | 0.60  | -        |
| Base           | 0.59  | 0.64     |
| Emotion        | 0.63  | 0.65     |
| Random         | 0.49  | -        |

**Table 11.** Accuracy of classifiers for all three categories based on audio only.

|                | Audio | FT Audio |
|----------------|-------|----------|
| RNN            | 0.41  | -        |
| Eight Emotions | 0.35  | -        |
| FSFM           | 0.44  | -        |
| Base           | 0.45  | 0.43     |
| Emotion        | 0.47  | 0.48     |
| Random         | 0.32  | -        |

### 5.3. Results on Combined Text and Audio Inputs

Our next set of experiments was designed to evaluate the accuracy of insulting and unsafe sentence detection when text and audio data were combined. It was assumed that adding audio with text would enhance the results of each of them separately, but not all the additions produced the expected results.

Tables 12–14 describe the experiment's results for various combinations of text and wave embedded vectors. The methods are compared to a baseline method that uses a randomly generated vector with a length of 768 sampled from the same distribution as the Wav2Vec vectors (referred to as the random vector), which was used instead of the Wav2Vec embedding vectors.

**Table 12.** Accuracy of insulting speech detection: text and audio.

| Audio Model | Audio Only | Audio and BERT | Audio and FT BERT | FT Audio and FT BERT |
|---|---|---|---|---|
| RNN | 0.60 | 0.75 | - | - |
| Eight Emotions | 0.48 | 0.79 | 0.87 | - |
| FSFM | 0.60 | 0.79 | 0.88 | - |
| Base | 0.61 | 0.75 | 0.87 | 0.87 |
| Emotion | 0.63 | 0.79 | 0.87 | 0.87 |
| Random | 0.51 | 0.75 | 0.86 | - |

**Table 13.** Accuracy of unsafe speech detection:text+audio.

| Audio Model | Audio Only | Audio + BERT | Audio and FT BERT | FT Audio and FT BERT |
|---|---|---|---|---|
| RNN | 0.60 | 0.76 | - | - |
| Eight Emotions | 0.50 | 0.76 | 0.83 | - |
| FSFM | 0.50 | 0.70 | 0.82 | - |
| Base | 0.59 | 0.75 | 0.83 | 0.84 |
| Emotion | 0.63 | 0.73 | 0.83 | 0.83 |
| Random | 0.49 | 0.75 | 0.83 | - |

**Table 14.** Accuracy of classifiers on all three categories based on speech.

| | | | | |
|---|---|---|---|---|
| RNN | 0.41 | 0.62 | - | - |
| Eight Emotions | 0.35 | 0.65 | 0.73 | - |
| FSFM | 0.44 | 0.59 | 0.73 | - |
| Base | 0.45 | 0.64 | 0.74 | 0.74 |
| Emotion | 0.47 | 0.64 | 0.74 | 0.74 |
| Random | 0.32 | 0.65 | 0.73 | - |

Addition of audio features to the text data was hypothesized to improve the accuracy of the results. When audio was combined with text, the following improvements were observed: the addition of audio vectors slightly improved the accuracy results with respect to the results based on the BERT source and eight-length random vector, but the accuracy of the results when using the fine-tuned audio vector did not improve that in almost all cases. We believe that only a relatively small improvement was observed because, in our context, almost all required information can be obtained from text, while audio signs cannot provide a clear indication of the situation. Another explanation for this phenomenon is that there are unsafe situations in which the offender will speak in a normal tone in an attempt to hide the danger. Similarly, insulting speech has no unique sound characteristics that distinguishes it from ordinary conversations. Interestingly, fine-tuning of the BERT vectors clearly improved the accuracy of the learning process, while fine-tuning of the audio vectors positively affected classification based on only audio, and exhibited no effect on text-and-audio-based classification. This was probably because the better audio fine-tuned embedding vector did not add new significant information, given the meaningful information extracted from the text.

## 6. Conclusions

In this study, we addressed the challenge of detecting insulting and unsafe situations using text and audio speech contents and using vector embedding for both text and audio,

with and without fine-tuning the embedding vectors. This challenge can be viewed as a classification problem and categorized into the following three classes: neutral sentences, sentences consisting of insulting content, and sentences indicating unsafe situations. We concentrated on situations relevant to children, particularly those with special needs. To solve this challenge and examine different classification schemes, we created a new dataset consisting of over 1981 Hebrew sentences, where for each sentence, both text and audio recordings were collected. We compared the classification accuracy of different state-of-the-art supervised and self-supervised methods. In particular, we used Heb-BERT embedding vectors for the text with and without fine-tuning and compared the accuracy of different ML methods running on the TF-IDF vectors with respect to their accuracy. For the audio content, we compared different variations of Wav2Vec embedding vectors, with and without fine-tuning, using different classical ML methods and DL methods.

In our experiments, we found that adding embedded wave information to the text information only slightly improved the overall accuracy. This may be because in unsafe situations, the main information conveyed was related to the words being spoken and not the way they are spoken or other characteristics of the audio. This may be especially true for a dialogue between people who are not close to family members.

This study makes significant contributions to two research areas. First, it can help develop automated agents to assist children with special needs. Second, it may advance the research on content classification and threat detection using text and voice signals. We would like to expand on these aspects.

There are some open issues that can be addressed in future studies. The first is the ability to recognize unsafe and insulting situations using only automated tools. Note that, in our study, Hebrew texts were produced manually based on audio. Observing whether audio could more significantly impact if an ASR was used to produce the text is interesting, given the fact that current Hebrew ASRs are not accurate. Another interesting expansion is to check for other possible unsafe situations. In this study, we collected a text and audio dataset concentrating on unsafe and insulting situations related to special needs children, and expanding the study to other unsafe situations such as violent situations between adults to detect domestic violence is interesting and important. Examining the effects of audio in different cultures and languages would also be interesting. In addition, examining whether adding video films and/or pictures of the examined events will increase detection accuracy is interesting, especially when considering unsafe situation detection.

**Author Contributions:** Conceptualization, M.A., A.A. and R.A.; Methodology, M.A., N.M., A.A. and R.A.; Software, M.A., N.M. and R.A.; Validation, N.M.; Writing—original draft, M.A. and N.M.; Writing—review & editing, A.A. and R.A.; Visualization, M.A. and N.M.; Supervision, A.A. and R.A.; Project administration, A.A. All authors have read and agreed to the published version of the manuscript.

**Funding:** This research was supported in part by the Ministry of Science, Technology & Space, Israel.

**Data Availability Statement:** The data presented in this study are available on request from the corresponding author. The data are not publicly available due to copyright of the sources used for composing the dataset.

**Conflicts of Interest:** The authors declare no conflict of interest.

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
