# Peer review of "Utilizing Machine Learning for Detecting Harmful Situations by Audio and Text"

_applsci, doi:10.3390/app13063927_

Round 1
Reviewer 1 Report
I thank the authors who, by submitting their work, allowed me to read this interesting article. Overall, the proposed method is very interesting: the reserach design is well described and the results seem promising.
I suggest to the authors some changes that could increase the quality of the paper.
Currently, the introduction is organized as an extended abstract: it provides context without going into detail about the research gaps, does not introduce a research objective aimed at overcoming the identified gaps, but briefly describes the research method and results.
I suggest that the authors rewrite the introduction to:
- provide a summary of the results of the state-of-the-art analysis they conducted, which were reported and discussed in detail in the "research background" session;
- highlight the current research gaps and define how their proposed approach can help reduce them.
Author Response
As suggested by the reviewer, we have added a summary of previous related work to the introduction and have further emphasized the current research gaps and how our approach solves them.
Reviewer 2 Report
The topic of this paper has important theoretical and application value and is well worth studying in depth. The paper uses machine learning models to fuse voice and video data sources to enter the automated research of recognition, and the research has certain academic contributions.
It is recommended to improve as follows:
(1) A case study can be combined to explain the background and significance of the research and why the integration of different types of data sources is carried out.
(2) Why use the BERT model for research? Explain the reasons theoretically. Or that the BERT model solves the problem of what aspects of the research problem are very good.
Author Response
(1) Thank you for the valuable offer to showcase the significance of utilizing diverse input sources. We have included a suitable demonstration in the third paragraph of the instructions.
(2) Thank you for your question. We believe that using the BERT model for research is advantageous for several reasons. BERT's ability to capture context and meaning is critical for understanding and analyzing natural language data, and its fine-tuning capabilities allow for high accuracy in analyzing specific datasets. By utilizing BERT for research, we can gain a more nuanced and accurate understanding of natural language data, which can help us identify patterns and trends that may be difficult to detect using other methods. We hope this clarifies why we have chosen to use the BERT model for our research. We added this motivation to Section 2.2.1.
Reviewer 3 Report
This paper describes a ML (machine learning) and DL (deep learning) approach to identify insulting text and sentences indicating unsafe conditions for children with special needs. The goal of the researchers is to ultimately develop an app that will identify these conditions for children with special needs.
The paper is very well written, clearly structured and easy to understand. The content is relevant both on the theoretical and practical level, I found the performance comparison of the different ML and DL approaches very useful.
The only small typo is on line 20, where in the sentence "These children may be involved in unsafe situations and discomfort situations, ___ being aware of it" "without" is missing.
Author Response
Thank you for your positive review. We put a lot of effort into our research and it's always gratifying to hear that it's making a difference. Your feedback and support are greatly appreciated and motivates us to continue producing high-quality work.
In addition, we fixed the typos on line 20.